# RLSF: Reinforcement Learning from Self-Feedback for Improved Logical Reasoning

## Abstract

Large Language Models (LLMs) have demonstrated impressive capabilities in generating coherent and contextually relevant text. These models arguably lack the ability to logically reason, an essential skill required to solving mathematical problems and programming tasks. While step-by-step prompting approaches show some promise, they often depend on finding a suitable prompt tailored to the specific model and task. In this work, we propose a simple, yet an effective approach to enhance reasoning capabilities by leveraging reinforcement learning (RL) and the confidence scores of a well-calibrated LLM. It involves optimising an implicit reward derived from the model's confidence levels in the answer to the reasoning task at hand. We generate preference data and fine-tune the LLM in a similar spirit to reinforcement learning from human feedback (RLHF), but without needing any human provided labels or preferences. Our results show that resulting reasoning abilities of an LLM improve and are transferable to other reasoning tasks. This warrants further investigation of RL as a facilitator for solving complex language tasks.

## 1 Introduction

Recent advances in large language models (LLMs) have led to impressive capabilities in text generation and comprehension (Brown et al., 2020; Ouyang et al., 2022). However, these models often struggle with tasks requiring deep logical reasoning, which is a critical limitation when deploying them in real-world applications such as legal analysis, scientific computation, and decision support systems (Kambhampati, 2024). While LLMs excel at generating contextually appropriate text, they frequently falter when asked to perform tasks that require maintaining consistency and accuracy across multiple reasoning steps.

To address this challenge, various techniques have been explored to enhance the logical reasoning capabilities of LLMs, the most promising being Chain-of-Thought (CoT) reasoning (Wei et al., 2022). In CoT, prompting encourages models to generate and articulate intermediate reasoning steps before arriving at a final conclusion. For example, prompts such as "Let us think step by step" or "Let us break it down" are used (Kojima et al., 2022). By making the reasoning process explicit, CoT reasoning improves the model's ability to handle complex tasks that require logical consistency and deep understanding. This approach, however, heavily relies on the prompt design, leading to inconsistent performance and user experience.

To address these limitations, approaches such as STaR (Zelikman et al., 2022) have demonstrated that models can be fine-tuned in a supervised manner to improve their logical reasoning capabilities. While this method can enhance the model's reasoning skills, it necessitates a large corpus of question-reasoning-answer triples, which is costly to obtain. Further, the fine-tuning process can be time-consuming and computationally expensive, limiting the scalability of the approach.

In contrast, Wang & Zhou (2024) found that while LLMs are often not able to provide a correct answer for a reasoning task using vanilla greedy decoding, the correct answer often appears within $K$ beams. Moreover, they found that confidence of answer tokens is correlated with the presence of reasoning in the decoding. Therefore, they propose generating a number of beams and selecting the one which has the highest confidence answer tokens. The results show that performance of LLMs on mathematical word problems improve, without the need for additional supervision or fine-tuning.

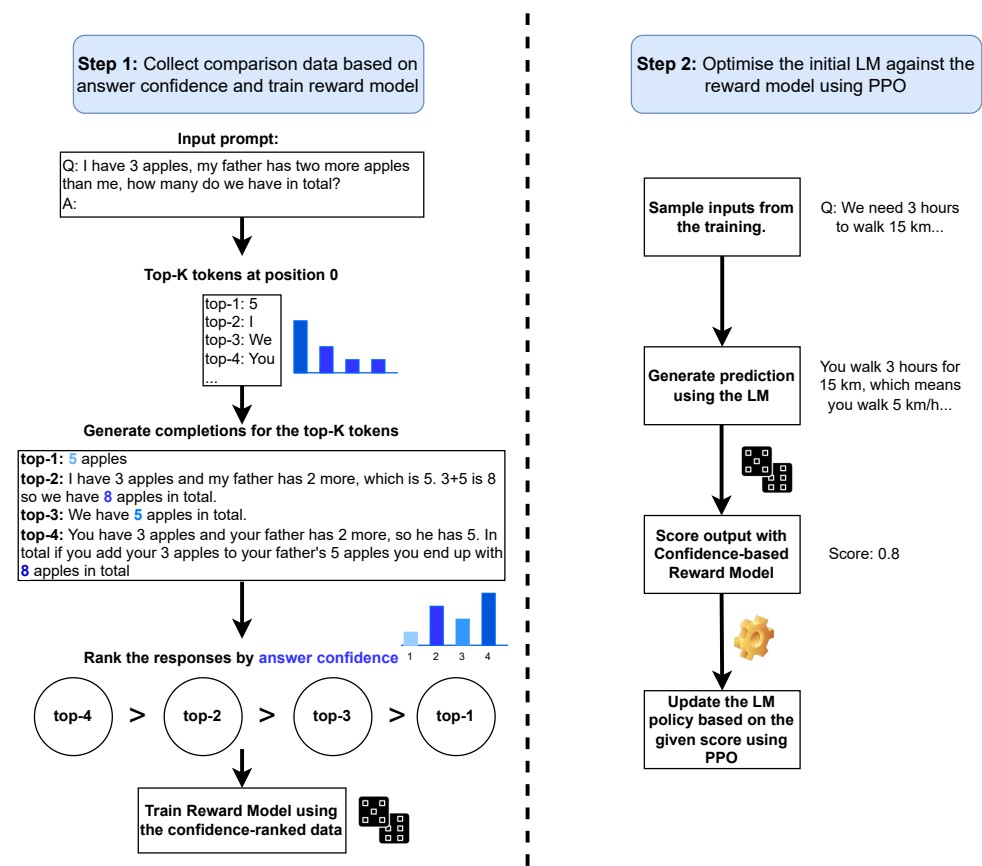

Figure 1: RLSF approach overview: For the top-K tokens at position 0 completions are generated, which are ranked based on answer confidence in descending order. The reward model is trained on the generated rankings, which is subsequently used to optimise the initial model using PPO. The example here is based on Wang & Zhou (2024).

However, this approach relies on generating multiple beams, which increases the computational cost of inference by an order of magnitude.

While these challenges have led to innovations within the domain of LLMs, it is important to note that reinforcement learning (RL) has long been a foundational framework for solving complex tasks that can be formulated as sequential decision making problems. The ability for RL algorithms to solve such tasks is evidenced by success in mastering games like Go, Chess, etc. (Silver et al., 2017).

In the domain of LLMs, RL has been deployed to reward the generation of the output which reflect human preferences, referred to as reinforcement learning from human feedback (RLHF; Christiano et al., 2017). This step of optimisation is often seen as critical to allow the models to achieve their impressive performance (Ouyang et al., 2022). Since RLHF is very sensitive towards the quality of the human provided preferences, Lee et al. (2023) propose that a superior LLM provides feedback for the training of smaller models. However this superior LLM still relies on the (accurate) human feedback.

Building upon the strengths of RL and addressing some limitations observed in CoT decoding, we propose a novel approach to reasoning tasks, which we name *Reinforcement Learning from Self Feedback (RLSF)*, illustrated in Figure 1. This approach is based on a simple observation: In a well-calibrated model, answer confidence is correlated with the presence of reasoning, which in turn leads to better quality answers. The generated beams can be ranked by answer confidence to

train a reward model that assesses both reasoning and answer quality. Then this reward model can be used to fine-tune the LLM via reinforcement learning.

We apply this simple idea to mathematical reasoning tasks, demonstrating that it significantly improves LLM performance on these tasks. Furthermore, we show that once fine-tuned, the resulting model also exhibits enhanced reasoning abilities across a broader range of tasks, even if the original model is not well calibrated on these tasks. Finally, while the training costs increase, the inference costs of the resulting LLM are equivalent to the vanilla LLM.

## 2 RELATED WORK

### 2.1 CONFIDENCE ESTIMATION IN LLMS

The reliability and calibration of confidence estimates in large language models (LLMs) are important in real-world applications. Overconfident models, which assign high confidence to incorrect answers, can undermine trust in these model. Well-calibrated confidence estimates are not only important for the model's trustworthiness, but also for improving its performance (Wang & Zhou, 2024).

Recent studies have analysed the calibration of confidence estimates across various LLM configurations. Supervisedly fine-tuned LLMs, trained on extensive datasets, have demonstrated well-calibrated token-level confidence (Kuhn et al., 2022; Xiao et al., 2022). However, aligned LLMs fine-tuned using RLHF often exhibit poorly calibrated token-level confidence (Tian et al., 2023; OpenAI et al., 2024). This discrepancy likely arises because RLHF optimizes for human preferences, which may not always correlate with the correctness of answers.

Traditional token-level confidence estimation methods primarily focus on the confidence of the final token in a response. This can lead to high confidence in the final token even if the answer is incorrect, as the model might generate a seemingly plausible continuation that does not reflect the correct answer. Recent advancements have shifted towards evaluating the confidence of the answer span, which has been shown to offer better calibration than the final or average token-level confidence (Kojima et al., 2022; Wang & Zhou, 2024).

### 2.2 CHAIN-OF-THOUGHT DECODING

The performance of large language models (LLMs) on reasoning tasks improves when the model generates a chain of thought (CoT). To elicit CoT reasoning, Wei et al. (2022) include examples of multistep reasoning in the prompt, while Kojima et al. (2022) prompt the model in a zero-shot manner to "think step by step". Reasoning capabilities can be further enhanced through specific training on CoT data (Chung et al., 2024), or by teaching the model how to reason (Zelikman et al., 2022).

CoT-decoding (Wang & Zhou, 2024) is proposed as a method which does not necessitate specific prompting or supervised fine-tuning. Instead, it elicits reasoning by exploiting the correlation between answer token confidence and the presence of CoTs. Multiple completions are sampled and the answer with the highest confidence is selected, as a confident answer is more likely to contain a CoT and, consequently, be correct. While this method effectively generates more accurate answers, it increases inference time by an order of magnitude due to the need to generate multiple beams for each input. CoT-decoding is mainly used for logical reasoning tasks, however in a recent study, Vukovic et al. (2024) employ it to improve generalisation of ontology relation extractors for task-oriented dialogue.

### 2.3 PREFERENCE LEARNING

In the field of large language models, RLHF is proposed for aligning models with human values and preferences (Christiano et al., 2017). In preference learning, LLMs are typically first fine-tuned on human-written question-answer pairs, followed by RLHF to further refine the model (Ouyang et al., 2022). During the RLHF phase, human annotators evaluate and rank sampled model outputs based on various criteria, such as helpfulness and safety. This ranking generates preference data that is

used to train a reward model, which in turn guides the model's further refinement to better adhere to the specified evaluation criteria and enhance answer quality.

Building on this foundational RLHF framework, Glaese et al. (2022) have introduced additional layers of control by instructing human workers to focus on specific rules, such as the avoidance of stereotypes. This refinement aims to improve the quality of the preference data, thereby enhancing the alignment process and mitigating the risk of adversarial attacks. Similarly, Bai et al. (2022) have employed a trained preference model to rank generated utterances, utilising this ranked data to train a reward model, thereby advancing the alignment and safety of the LLM. Despite these advancements, all existing RLHF approaches are heavily dependent on human labellers for generating preference and safety data.

## 3 METHODOLOGY

Our method is based on the observation that if the model is well calibrated the confidence of the answer correlates with the presence of reasoning and hence with the accuracy of this answer. Therefore, a sequential-decision making process is needed that chooses the tokens of the generated text in such a way that the confidence of the answer tokens is maximal. This can be achieved via reinforcement learning.

### 3.1 CHAIN-OF-THOUGHT DECODING PRELIMINARIES

The goal of this method is to extract the inherent reasoning ability of a large language model by generating multiple hypotheses for a given input q. In contrast to traditional beam-search methods, in CoT decoding one samples the $K$ highest-probability tokens $t_i, i = 1, \cdots, K$ at the very first decoding step. From here, each hypothesis $h_i, i = 1, \cdots, K$ is expanded using standard autoregressive decoding, i.e. $h_i[0] = t_i$. Then the tokens contained within the answer $a_i, i = 1, \cdots, K$ to the question, q, are identified in each hypothesis $h_i$. This is done by appending the text "So the answer is" to each hypothesis $h_i$, subsequently continuing the decoding process to generate $a'_i, i = 1, \cdots, K$. Next, the repeated answer span is located in the hypothesis $h_i$ using string matching to obtain $a_i$.

The final beam a is the one with the highest answer confidence of the model, $a = \arg\max_i \mathbb{d}(h_i)$. The confidence in the answer is calculated as the average token level probability disparity of the answer tokens (Wang & Zhou, 2024). That is, given an answer a, consisting of $M$ tokens $t_m, m = 1, \cdots, M$, the confidence of the model in the answer is calculated as:

$$\mathbb{d}(a) = \frac{1}{M} \sum_{m=1}^{M} \mathsf{p}\left(t_j^{(1)} \middle| h_{<j}, \mathsf{q}\right) - \mathsf{p}\left(t_j^{(2)} \middle| h_{<j}, \mathsf{q}\right).$$

Here, $\mathsf{p}\left(t_j^{(1)} \middle| h_{<j}, \mathsf{q}\right)$ is the probability of the most likely token at position $j$ in the hypothesis, and $\mathsf{p}\left(t_j^{(2)} \middle| h_{<j}, \mathsf{q}\right)$ is the probability of the second most likely token at position $j$ in the hypothesis.

Probability disparity tends to be a more reliable indicator of the model's confidence than the probability of the token itself. This is because it also considers the probability of the second most likely token, which helps capture the spread of probability mass across the vocabulary. Higher disparity corresponds to the model having a high certainty in sampling of the answer tokens, since an alternative continuation is significantly less likely to be sampled.

To further improve the reliability of the confidence scores, we consider the frequency of unique answers in the generated hypotheses. It has been shown empirically that the answer is more likely to be correct if it is the most frequent answer in the set of hypotheses (Wang & Zhou, 2024). To combine confidence and frequency, we calculate the final confidence score for an answer as the sum of the confidence scores of the beams that contain that answer.

It is important to note that CoT decoding does not guarantee reasoning (Wang & Zhou, 2024), but rather builds on the observed correlation of answer confidence and reasoning to improve answers. The ability to generate rationales is likely learned during pre-training, since logical reasoning data often contains reasoning. However, LLMs tend not to reason when prompted directly and output

the answer directly instead. Our approach aims at bringing the inherent LLM reasoning abilities forward to be predicted via greedy decoding.

## 3.2 REINFORCEMENT LEARNING FROM SELF FEEDBACK

Reinforcement learning (RL) relies on a reward signal to guide the learning process. Thus, defining the reward appropriately is an important aspect of any RL-based approach.

In our approach, we utilise a reward model, inspired by Reinforcement Learning from Human Feedback (RLHF; Christiano et al., 2017). In RLHF, the reward model is a large language models which is trained using a combination of pre-training, supervised fine-tuning, and fine-tuning on human preference data. The human preference data is used to train a reward model, which assigns a score to generated sequences based on human judgements. The objective function for training the reward model is given as follows (Bradley & Terry, 1952):

$$\mathcal{L}\left(h^{(1)}, h^{(2)}; \boldsymbol{\theta}^{\text{rew}}\right) = -\log\left(\sigma\left(R\left(h^{(1)}\right) - R\left(h^{(2)}\right)\right)\right),$$

where hypothesis $h^{(1)}$ is preferred over hypothesis $h^{(2)}$, $\sigma(\cdot)$ is the sigmoid activation function, and $R(\cdot)$ is the reward assigned to a hypothesis by the reward model parameterised by parameters $\boldsymbol{\theta}^{\text{rew}}$.

Once the reward model is trained, it can be used to assign reward values to the generated sequences. In the context of reinforcement learning, this implies that each token in a sequence is assigned a reward of zero, except for the final token, which receives the reward predicted by the model for the entire sequence.

In contrast to approaches that rely on human preferences for ranking, our method ranks hypotheses based solely on the confidence scores of their predicted answers. These confidence-based rankings are then used to train the reward model, and the correctness of the answers is not explicitly considered when ranking the hypotheses. Given that the model relies entirely on its own confidence scores rather than external feedback, we refer to this approach as Reinforcement Learning from Self-Feedback (RLSF).

## 4 EXPERIMENTAL SETUP

### 4.1 DATASETS

To evaluate the efficacy of RLSF, we conduct experiments using logical reasoning datasets. Specifically, we utilise the Multi-Arith and GSM8K (Cobbe et al., 2021) mathematical word problem datasets during the training phase.

Additionally, we evaluate the method on the StrategyQA commonsense question answering dataset (Geva et al., 2021), as well as on synthetic reasoning tasks, such as tracking the state of a coin through a sequence of actions and concatenating the first or last letters of words, which was used in Khot et al. (2022).

### 4.2 CALIBRATION

As outlined in Section 3, our approach relies on the calibration of the language model. For evaluating the calibration, we use the estimated calibration error (ECE; Naeini et al., 2015) that measures the alignment between model confidence and prediction accuracy. In other words, ECE measures how much model confidence tells us about answer quality; the lower ECE the better. Consequently, we decided to use the Phi-2 model (Hughes, 2023) in all our experiments, as it demonstrated superior confidence calibration compared to alternative models, including Mistral (Jiang et al., 2023) and Gemma (Team et al., 2024) (see Table 1). Furthermore, all fine-tuning, including the training of the reward model, was performed using Low-Rank Adaptation (LoRA) (Hu et al., 2021).

### 4.3 REWARD MODEL

We train a reward model based on the self-feedback mechanism described in Section 3.2. For our reward model, we use an instance of the Phi-2 model, with a reward prediction head fine-tuned using

| Model | Multi-Arith | | GSM8K | |
|---|---|---|---|---|
| | ECE | Accuracy | ECE | Accuracy |
| Phi-2 | 11.9 | 60.6 | 48.8 | 42.7 |
| Mistral 4b Instruct | 24.3 | 46.3 | 53.1 | 37.6 |
| Gemma 2b Instruct | 62.5 | 29.7 | 84.8 | 11.3 |

Table 1: Confidence calibration of different language models on the mathematical reasoning tasks, together with the accuracy scores of Chain-of-Thought decoding using 10 beams.

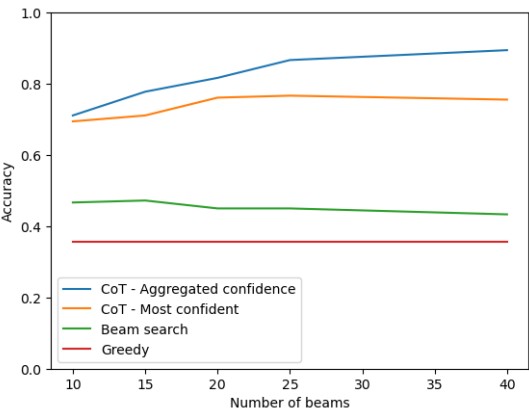

Figure 2: Comparison of decoding strategies on the multi arithmetic task using the Phi2 model.

LoRA. To train this reward model, a preference dataset is required, which we generate using questions from the Multi-Arith and GSM8K mathematical reasoning datasets. For the construction of our preference dataset, we do not accumulate answer confidences, as we wish to maintain the individual hypotheses. Consequently, we generate 10 hypotheses per input, as Figure 2 demonstrates no significant improvement in CoT decoding performance without accumulation beyond 10 hypotheses. To ensure stability during learning, the rewards are re-scaled so they are in interval $[0, 1]$. We train both the reward model and the prediction model using the transformer reinforcement learning library (TRL; von Werra et al., 2020).

## 5 RESULTS AND DISCUSSION

### 5.1 CHAIN OF THOUGHT DECODING

In this section, we evaluate the original CoT decoding strategy (Section 3.1) on the multi-arithmetic mathematical reasoning task (Figure 2). Our results confirm that it outperforms both greedy decoding and beam search. We observe that greedy decoding achieves the lowest performance, as anticipated. Beam search performs better than greedy decoding, however, interestingly, its performance declines as the number of beams increases. We hypothesise that this decline is due to beam search's reliance on token-level probabilities to prune beams. In contrast, CoT decoding, which utilises answer-level confidence to rank beams, significantly outperforms both greedy decoding and beam search. This performance is further enhanced when answer confidences are accumulated.

### 5.2 MATHEMATICAL REASONING

In this section, we evaluate the proposed RLSF approach on the Multi-Arith and GSM8K mathematical reasoning tasks. We fine-tune the Phi-2 model using the RLSF approach, incorporating the reward model obtained in Section 4.3 and the proximal policy optimisation (PPO) algorithm (Schulman et al., 2017). As an alternative to the RLHF framework we also train the model using direct preference optimisation (DPO; Rafailov et al., 2024) that does not need a separately trained reward model.

| | |
|---|---|
| **Question**: | Faye had 34 coloring books. If she gave away 3 of them, but then bought 48 more, how many would she have total? |
| **Answer**: | **79** |
| **Greedy Decoding**: | Faye would have **85** coloring books in total. |
| **CoT Decoding** (20): | Fae has a total of 65 coloring books (34 original books + 48 purchased books - 3 given away books = **65**). |
| **RLSF**: | Faye had 34 coloring books. She gave away 3 of them, so she had $34 - 3 = 31$ coloring books left. She then bought 48 more, so she had $31 + 48 = $ **79** coloring books total. |

Figure 3: Comparison of answers generated by the Phi-2 model using greedy decoding, Chain-of-Thought (CoT) decoding with 20 beams, and the proposed RLSF approach on a question from the Multi-Arith dataset.

In Table 2, we observe that the model fine-tuned with the RLSF approach outperforms the base model, when using greedy decoding, on both the Multi-Arith and GSM8K mathematical reasoning tasks. Additionally, this model either surpasses or matches the performance of the base model when using CoT decoding with up to 25 hypotheses and accumulating answer confidences. This demonstrates that through the self-reflection process, the model learns to generate more accurate responses without the need for beam search or other decoding strategies. The comparison with DPO that does not utilise a dedicated reward model further illustrates the power of the reward model, although the DPO-trained model still outperforms the greedy baseline.

Moreover, in Figure 3, we present a scenario where a direct answer obtained via greedy decoding fails to answer the question correctly. In this example, CoT decoding also fails to provide the correct answer, offering an explanation for a possible answer rather than reasoning through to the correct conclusion. Finally, our proposed RLSF approach successfully breaks the problem into smaller, manageable tasks and ultimately arrives at the correct answer.

| Method | Number of Beams | Multi-Arith | GSM8K | Decoding Cost |
|---|---|---|---|---|
| Greedy Decoding | 1 | 34.4 | 25.3 | $\mathcal{O}\left(n^2\right)$ |
| | 10 | 60.6 | **42.7** | $\mathcal{O}\left(Kn^2\right)$ |
| CoT Decoding | 25 | 76.7 | 39.6 | $\mathcal{O}\left(Kn^2\right)$ |
| | 25 (agg) | **86.7** | 40.8 | $\mathcal{O}\left(Kn^2\right)$ |
| RLSF | 1 | 78.9 | **42.7** | $\mathcal{O}\left(n^2\right)$ |
| DPO SF | 1 | 56.7 | 35.3 | $\mathcal{O}\left(n^2\right)$ |

Table 2: Phi-2 accuracy results for Multi-Arith and GSM8K. The decoding cost is represented as the computational complexity of the decoding process where the context and generated tokens amount to $n$, represented using $\mathcal{O}$ notation and K is the number of beams in CoT decoding. Best Results in each dataset are highlighted in **bold** and second best are underlined.

## 5.3 GENERALISATION

| Method | Coin Flip | StrategyQA | First Letter | Last Letter |
|---|---|---|---|---|
| Greedy Decoding | 68.7 | **67.5** | 66.7 | 7.0 |
| CoT Decoding (10 Beams) | 23.8 | 48.4 | 50.7 | 6.0 |
| RLSF | **76.0** | 65.5 | **74.7** | **19.3** |

Table 3: Performances of tasks not seen during RLSF training measured in accuracy. Best results for each dataset in **bold**.

The natural follow-up question is whether the above results generalize to other tasks. To explore this, we evaluate the model from Section 5.2 on the StrategyQA commonsense question answering dataset, as well as on synthetic reasoning tasks, such as tracking the state of a coin through a sequence of actions and concatenating the first or last letters of words.

In Table 3, we observe that the RLSF model performs at least as well as the base model on the StrategyQA dataset, while significantly outperforming the base model on the synthetic reasoning tasks. Unlike the CoT decoding approach, which often led to reduced performance on these tasks, the RLSF approach consistently enhances performance across all evaluated tasks. This consistent improvement demonstrates the RLSF model's generalisation ability beyond mathematical reasoning, extending to other logical reasoning tasks.

We hypothesise that the reduction in performance observed with CoT decoding is due to the difficulty in correctly locating the answer span in these more complex tasks. This issue arises because CoT decoding relies heavily on re-ranking hypotheses based on the answer-span confidence. When the correct answer span is challenging to identify, this can lead to undesirable behaviour during the re-ranking process, with incorrect spans being ranked more favourably. In the case of coin flip and last letter the problem arises from the answer tokens being mentioned several times in the generated reasoning, decreasing the impact of the final answer mention when choosing the best beam.

In contrast, the RLSF model avoids this problem, as it does not depend on identifying the answer span during inference; only the final answer needs to be parsed for evaluation. This allows the RLSF model to maintain its high performance, even in tasks where span identification is inherently difficult. As a result, the RLSF model remains robust across different types of tasks, showcasing its versatility and ability to adapt to varying reasoning challenges without the limitations imposed by answer span-dependent methods.

Interestingly, we found that the calibration of the RLSF-trained model is improved compared to the baseline on GSM8K and the synthetic last letter concatenation task, as seen in Table 4.

| Model | Multi-Arith | | GSM8K | | Last Letter | |
| | ECE ↓ | Accuracy ↑ | ECE ↓ | Accuracy ↑ | ECE ↓ | Accuracy ↑ |
|---|---|---|---|---|---|---|
| Phi-2 | 11.9 | 34.4 | 48.8 | 25.3 | 21.3 | 7.0 |
| Phi-2 RLSF | 16.5 | 78.9 | 41.4 | 42.7 | 17.8 | 19.3 |

Table 4: Confidence calibration and "Greedy" Decoding accuracy of the Phi-2 model before and after RLSF.

## 5.4 REWARD MODEL QUALITATIVE ANALYSIS

In Figure 4, we illustrate the rewards learned by the reward model alongside corresponding example answers. We observe that the reward model generally assigns higher rewards to answers that exhibit more logical reasoning, and thus tend to be more accurate. However, it is worth noting that the reward model does not always assign the highest reward to the most accurate answer. We observe instances where the reward model assigns a higher reward to a less accurate answer, both with and without logical reasoning.

## 5.5 DISCUSSION

**Does the RLSF model reason better than the base model?** Overall, the RLSF model exhibits enhanced reasoning capabilities compared to the base model, as evidenced by its ability to produce more detailed and accurate responses. The increased length and quality of answers suggest that the model engages in deeper reasoning. However, the reasoning process is not without flaws. In some cases, the model generates a final answer prematurely and then attempts to justify it retroactively, which indicates that the answer is not always the result of a coherent reasoning process. Additionally, there are instances where the model digresses, offering irrelevant or repetitive explanations without contributing meaningful logical reasoning. Thus, while the RLSF model demonstrates improved reasoning, there remain areas where its reasoning could be further refined to achieve greater precision and coherence.

**Is the RLSF approach more computationally efficient than CoT decoding?** While the RLSF approach requires comparable computational resources to CoT decoding during the creation of the preference dataset, it also incurs additional costs for training the reward model and fine-tuning the language model. However, during inference, the RLSF approach is as computationally efficient as

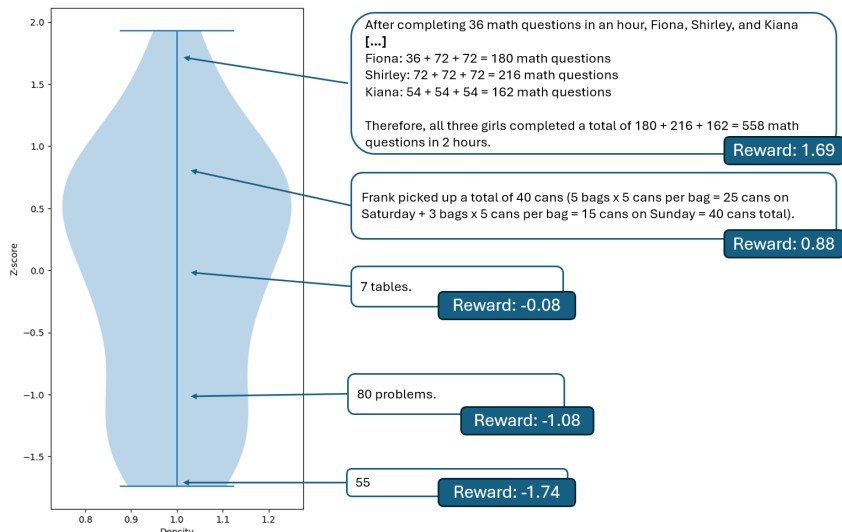

Figure 4: Illustration of the rewards learned by the reward model. On the y-axis is the assigned reward and on the x-axis the density, i.e. how many responses get assigned a specific reward.

the base model using greedy decoding, as it does not require additional decoding steps. Thus, for the specific mathematical reasoning tasks in this study, the RLSF approach may be considered more computationally intensive than CoT decoding. Nevertheless, the generalisation capability of the RLSF approach, as demonstrated in Section 5.3, suggests that when applied to a broader range of tasks, the RLSF method offers greater computational efficiency compared to CoT decoding, especially as it eliminates the need for multiple hypotheses or complex decoding strategies.

## 5.6 LIMITATIONS

An important limitation of our approach is the fact that it only works when the calibration of the LLM is good enough. For evaluating the calibration of an LM with ECE labelled data is needed and it only gives us an idea of the calibration for the specific task.
While the reliance on the identification of the answer span can be considered a bottleneck, it can be argued that is needed for evaluating the performance of an LM on logical reasoning tasks, so this problem is not limited to our approach in particular. The correlation-based heuristic for ranking the responses does not ensure reasoning and answer quality in all cases. It is also important to mention that the initial model has to be capable of generating reasoning in some considered beams for the method to improve reasoning capabilities.

## 6 CONCLUSION

In this work, we proposed a novel approach, Reinforcement Learning with Self-Feedback (RLSF), to improve reasoning capabilities in language models. Our method leverages self-reflection during training, coupled with a reward model that guides the language model to generate more accurate and logically consistent responses. Through evaluation on mathematical reasoning tasks, such as Multi-Arith and GSM8K, as well as common-sense reasoning and synthetic tasks like StrategyQA and coin-state tracking, we demonstrated that the RLSF approach significantly outperforms baseline models and decoding techniques, including CoT decoding.

Our contributions include:

1. the introduction of the RLSF framework, which improves a model's reasoning ability by utilising its own feedback in a reinforcement learning loop,

2. the development of a reward model trained from model-generated preferences, and

3. empirical evidence showing that RLSF outperforms established methods across multiple reasoning domains.

Despite these advancements, there are still areas for future improvement. In particular, we observed that the model's reasoning can be imperfect in certain cases, with tendencies to justify answers post hoc or produce irrelevant explanations. A more critical limitation of the current RLSF framework is that it performs single-step reasoning, which restricts its ability to handle tasks requiring long-term planning and more complex decision-making. While reinforcement learning is leveraged to enhance reasoning, RLSF, like many RLHF-based approaches, remains focused on single-turn interactions. The incorporation of long-term planning into the reinforcement learning process for language models is still in its early stages (Zhou et al., 2024).

In future work, we envision addressing these limitations by integrating more advanced reasoning mechanisms, particularly multistep reasoning and long-term planning, into the RLSF framework. By extending beyond single-turn optimisation, models could potentially reason over extended sequences of actions, making them more suitable for tasks that require complex reasoning or decision-making.

## 7 ETHICS STATEMENT

Although the presented approach improves performance and encourages reasoning in LLMs, it does not include measures to prevent harmful output. Depending on the calibration of the LLM and the underlying data used for comparison data collection, it might be possible that harmful behaviour of the LLM is reinforced.
However, by reducing the need for human intervention for reasoning training of LLMs, there are potentially more resources available to mitigate harmful LLM behaviour. Apart from that, the learned reasoning could possibly reduce the risk of harmful behaviour, since the LLMs learn not to rely on the highest probability response from the pre-training data distribution.

## 8 REPRODUCIBILITY STATEMENT

The datasets utilised in our experiments are all publicly available. Furthermore, the trained Phi-2 model is an open source model, facilitating reproducibility. Upon publication, all the code used for our experiments will be made publicly available, and the git repository will be linked in the final version of the paper. The main sections of the paper needed for reproducing the experiments are Sections 3 and 4 where the method for obtaining the beams, computing the answer confidence and ranking the responses, and the datasets are described.

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

# A APPENDIX

In Figure 5 we see the reliability diagram of the RLSF-trained model and the baseline on the Coin-flip task. Although increasing the ECE, the calibration diagram is better aligned to the diagonal after RLSF training.

## A.1 RLSF-MODEL RELIABILITY DIAGRAM

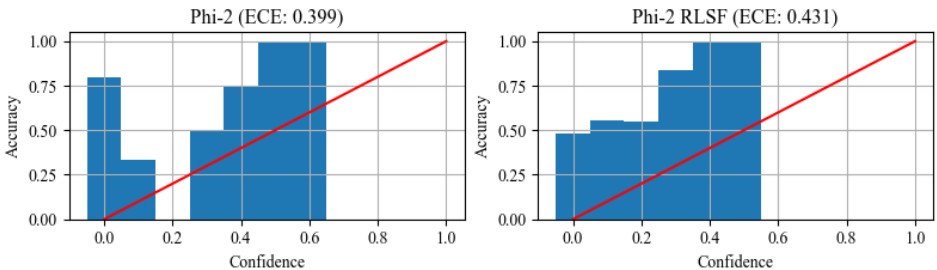

Figure 5: Reliability diagrams of the Phi-2 model before and after RLSF on the Coin-flip task.

