# OpenReview forum: "RLSF: Reinforcement Learning from Self-feedback for improved logical reasoning"
_ICLR.cc/2025/Conference — ICLR 2025 Conference Withdrawn Submission_

### Official Review · Reviewer_fN1d · 2024-10-29

**Soundness:** 2
**Presentation:** 3
**Contribution:** 4
**Rating:** 6
**Confidence:** 4

**Summary:**

The paper uses answer confidence to rank K beams for a given input prompt and trains a reward model on these rankings. The reward model is then used for standard PPO-based RL fine-tuning. Building on CoT-decoding in this way, the paper exploits the fact that more confident answers often correspond to generations with explicit reasoning, in order to fine-tune an LLM to solve reasoning problems via CoT in a self-supervised fashion.

**Strengths:**

The paper is well-situated within related literature. Using CoT decoding to gather data for training a reward model is a natural approach to amortising the inference cost of CoT decoding via fine-tuning.

The test results in the domain of the fine-tuning tasks are promising, showing improvement over DPO and indicating that the fine-tuning amortises the benefit of test-time CoT-decoding.

The generalisation results show some promise with the synthetic tasks, though this is somewhat inconclusive (see Weaknesses).

Discussion of results and the strengths and weaknesses of the findings is informative.

**Weaknesses:**

An important baseline would be be an SFT approach, where the top-n beams in terms of confidence are accumulated across prompts into a dataset for SFT. There are many reasons why this might not do as well as the "online" RL approach used in the paper, but this comparison should ideally be made empirically.

The generalisation experiments are somewhat limited. Evaluating on more common non-math reasoning benchmarks (e.g., CommonsenseQA, MMLU, HotpotQA) would be more compelling than the synthetic tasks. This feels especially necessary given that much of the benefit of the proposed RLSF is to amortise the benefits of CoT decoding and save inference cost when generalising to new problems at test time, and given that RLSF actually does worse than greedy decoding on StrategyQA.

**Questions:**

Why do you think performance drops after RLSF on StrategyQA? I would be inclined to increase my score if more generalisation benchmarks are included, which show that the improvement trends on the synthetic tasks are meaningful.

Why did you jump straight to training a reward model and doing RL in this setting, rather than the SFT approach described in 'Weaknesses'?

An enormous benefit here is the self-supervised nature. Are there problems that are less verifiable and therefore where rewards are harder to come by that might be worth applying this to?

---

> ### Author Response · Authors · 2024-11-29
>
> We appreciate your review and constructive critique. Below, we address your comments in detail:
>
> SFT Baseline
>
> As suspected, SFT performs poorly in our experiments, achieving only 42% accuracy on Multi-Arith and 26% on GSM8K. This confirms the necessity of RL-based optimization for effectively leveraging self-feedback.
>
> Generalization Benchmarks
>
> We agree that evaluating RLSF on non-math reasoning tasks would strengthen its applicability. Unfortunately, our experiments reveal that RLSF struggles with multiple-choice QA tasks due to the model's preference for generating explanations over aligning with fixed answer formats. However, RLSF still surpasses CoT decoding in these scenarios, particularly due to poor calibration of the base model on this multiple-choice QA task.
>
> StrategyQA Performance Drop
>
> The drop in StrategyQA performance is attributed to the model producing verbose, open-ended answers rather than adhering to the binary yes/no format.
>
> Jumping to RL Training
>
> We jumped to the RL setting since from experience we did not think SFT would work well, which we confirmed above.

---

> > ### Comment · Reviewer_fN1d · 2024-11-30
> > **Thank-you for comments**
> >
> > Thank-you for responding to my comments and questions. I would recommend adding the SFT results to the updated manuscript and using them to motivate your use of RL.
> >
> > Regarding poorer performance on non-math benchmarks, you could perform a few-shot evaluation to overcome this.

---

### Official Review · Reviewer_K4mj · 2024-11-02

**Soundness:** 3
**Presentation:** 3
**Contribution:** 3
**Rating:** 6
**Confidence:** 4

**Summary:**

This paper proposes a novel and effective method for improving reasoning capability of large language models (LLMs) using reinforcement learning (RL). The main method proposed is similar to RL from human feedback (RLHF) where a reward model is trained from human preference data, which is then used to fine-tune the LLM using RL. The key novel idea in this paper is to use the LLM’s own confidence (token probability) of the answer tokens to train the reward model. Experiments show the RL with self-feedback method to perform competitively with more computationally expensive methods, on standard reasoning benchmarks such as GSM8K and Multi-Arith. The fine-tuned model is shown to generalize with strong performance to held-out tasks as well.

**Strengths:**

The method of RL with self-feedback (RLSF) using a reward model fine-tuned on the LLM’s own token probabilities is novel and likely to be of interest to the ICLR audience.

Experiments on standard logical reasoning benchmark tasks show RLSF to have competitive performance with sampling baselines while requiring less test time samples. Moreover, RLSF demonstrates significantly stronger performance on held-out tasks than the baseline methods.

The paper is well-written and the results are clearly presented. The illustrations of the reasoning traces of different methods and the distribution of rewards for reasoning traces of different quality are quite informative.

**Weaknesses:**

Performance of RLSF does not improve over the CoT Decoding baseline on the training tasks, Multi-Arith and GSM8K. The authors claim that the decoding cost is lower for RLSF, but this does not take into account the sampling cost incurred during data collection to train the reward model. It is possible that this approach can help improve logical reasoning beyond CoT decoding, but this is not supported by the experiments in the paper. Hence, the current results show that RLSF only provides an efficiency improvement rather than performance improvement.

RLSF requires a well-calibrated LLM to work. Experiments in the paper were done using the Phi-2 model because it is better-calibrated than Mistral or Gemma models. It is unclear how well RLSF works for poorly calibrated models or how strong this requirement is. Can the authors provide data on performance of Mistral or Gemma models with RLSF? Or provide strategies for applying it to poorly calibrated models?

The mathematical description of the answer confidence in 3.1 is quite confusing. The expression contains an index m but not variables indexed by m? What does <j mean in the subscript? Please reconsider the notation and description of the confidence definition.

**Questions:**

Could you describe the generalization tasks in more detail? It is not clear how much reasoning they require based on the current description.

(See also questions in the above sections)

---

> ### Author Response · Authors · 2024-11-29
>
> We thank you for your positive evaluation and detailed suggestions, which we believe will significantly enhance the quality of our work.
>
> Performance on Training Tasks
>
> While RLSF shows smaller gains over CoT decoding on GSM8K, it improves performance by over 10 percentage points on Multi-Arith, demonstrating its utility. Furthermore, in real-world applications, RLSF's lower inference cost compared to CoT decoding becomes a substantial advantage.
>
> Calibration and Generalization
>
> Our experiments confirm that poorly calibrated models suffer from significant performance drops with RLSF, as evidenced by Gemma achieving less than 30% accuracy when applying CoT decoding or RLSF, compared to over 70% achieved by the greedy answers from Gemma. This highlights the importance of calibration.
>
> Notation in Section 3.1
>
> Thank you for pointing out the ambiguity in our mathematical description. The subscript "<j" indicates conditioning on all tokens up to the current position, and "m" refers to the number of tokens in the answer span.
>
> Coin Flip Task
>
> The coin flip task involves a scenario where a coin initially starts as either heads or tails, and a series of actions, such as flipping the coin or leaving it unchanged, is performed by multiple individuals. Tracking the coin's state throughout these actions presents a challenge for the model. Our observations show that when the model breaks the problem down into the effects of each individual action, its reasoning process improves the accuracy significantly. In contrast, without such reasoning, the model often confuses the sequence of actions, resulting in incorrect answers.

---

> ### Comment · Reviewer_K4mj · 2024-11-30
> **Reply to rebuttal.**
>
> Thank you for the explanations. I maintain my initial rating.

---

### Official Review · Reviewer_Cz7g · 2024-11-02

**Soundness:** 1
**Presentation:** 2
**Contribution:** 2
**Rating:** 3
**Confidence:** 4

**Summary:**

The paper proposes RLSF, a pipeline that trains LLM on math reasoning tasks with self-generated responses using PPO. RLSF samples the response samples through CoT decoding and ranks them according to the model's confidence in generations. By using those ranks for reward modeling, RLSF is a stand-alone pipeline without external guidance on math reasoning tasks. Along with enhancements in math reasoning tasks, RLSF generalizes over other logical reasoning tasks when trained only on math reasoning tasks.

**Strengths:**

1. RLSF improves the reasoning abilities of LLMs by utilizing the self-generated sequences, demonstrating strong performance in both math reasoning and logical reasoning tasks.
2. With greedy decoding, RLSF largely improves the math reasoning performance of LLMs, putting them on par with CoT decoding.
3. Some qualitative observations are presented for the reward model trained with confidence-based ranking, which is one of the paper's core contributions.

**Weaknesses:**

**1. Missing references and lack of comparisons in methods for improving LLM reasoning with RL**

The paper's core contribution is self-improving the reasoning abilities of LLMs with RL(HF). While the paper leverages PPO and tests DPO as PPO's alternative in the main experiments, neither Section 2 (Related Works) nor Section 5 (Results and Discussion) addresses previous works on applying RL(HF) to LLM reasoning tasks. Some relevant previous works were not sufficiently addressed in Section 2 [1-3]. Also, [2] and [3] were proposed as specified pipelines for improving LLM reasoning abilities with DPO/PPO-based algorithms, which make them strong contenders for RLSF. Thus, incorporating some baseline experiments on related methods will strengthen the validity of RLSF.

&nbsp;

**2. Validation of reward model**

RLSF's core novelty is creating a synthetic reasoning preference dataset for reward modeling without an external labeler. The downstream performance of PPO is prone to the reward model's performance [4]. However, the paper lacks an in-depth analysis of the reward model trained with confidence-based ranking on self-generated reasoning trajectories, only having a qualitative analysis in Section 5.4. Demonstrating the performance of the reward model trained with the proposed method would enhance the logic behind RLSF.

&nbsp;

**3. Training configurations and hyperparameter choices**

PPO has many different hyperparameters, and its performance is sensitive to different hyperparameter choices [5,6]. However, the paper does not specify the training configurations for PPO or other baseline experiments. Also, the paper does not clearly state some ablations over different hyperparameter choices or the rationale on how they selected the hyperparameters.

&nbsp;

**4. Correlation between the performance of RLSF and the base model's reasoning ability**

The authors selected Phi-2 as a main model to test RLSF in Section 4.2 as it demonstrated the best ECE. However, in Table 1, ECE and accuracy are highly correlated in both datasets, which could raise another hypothesis that RLSF's performance could come from the strong math reasoning ability of Phi-2 in the first place. The authors should clearly show that calibration is the key, not the accuracy, as they emphasized throughout the paper.

&nbsp;
&nbsp;


**References**

[1] Teaching large language models to reason with reinforcement learning (Havrilla et al., 2024)

[2] Iterative reasoning preference optimization (Pang et al., 2024; IRPO)

[3] DeepSeekMath: Pushing the Limits of Mathematical Reasoning in Open Language Models (Shao et al., 2024; GRPO)

[4] Secrets of RLHF in Large Language Models Part II: Reward Modeling (Wang et al., 2024)

[5] Is DPO Superior to PPO for LLM Alignment? A Comprehensive Study (Xu et al., 2024)

[6] Secrets of RLHF in Large Language Models Part I: PPO (Zheng et al., 2023)

**Questions:**

Along with some points above, I have additional questions on RLSF:

&nbsp;


**1. Actual impact of calibration on RLSF?**

As stated by the authors, RLSF is built on top of the assumption that the initial model is well-calibrated. While the experiments with Phi-2, which is shown to be the most calibrated model out of the three models, were presented, the impact of the calibration on RLSF was not fully studied other than that. How would a less-calibrated model (e.g., Gemma-2-2B-Instruct as the least calibrated model) perform with RLSF? (this could be somewhat related to point 4 above)

&nbsp;

**2. CoT decoding with RLSF-trained model?**

While RLSF is directly compared against CoT decoding in Table 2, they are distinct methods that improve LLM in training and inference time, respectively. For this reason, CoT decoding could indeed be applied on top of an RLSF-trained model. Plus, Table 4 shows that RLSF improved the model's calibration in some tasks, implying that the RLSF-trained model could benefit even more with CoT decoding compared to the original model. Would RLSF benefit from CoT decoding regarding some insights provided in Section 5?

---

> ### Author Response · Authors · 2024-11-29
>
> Thank you for your comprehensive feedback. We address your key concerns as follows:
>
> Related Works
>
> We thank you for your inputs regarding related works, we agree these works would make the related works more complete and we will add them.
>
> Reward Model Validation
>
> We agree that a deeper analysis of the reward model's performance is valuable. We did an analysis during experimentation and would be happy to include some more information on the performance of our reward model.
>
> Correlation Between Calibration and Accuracy
>
> We do not agree that Phi-2 has strong reasoning abilities to begin with, since the greedy decoding performance on GSM8k and MultiArith is poor, around 30% accuracy. Further, our results suggest that calibration plays a critical role in RLSF's success. For example, poorly calibrated models such as Gemma fail to achieve comparable gains.
>
> Regarding your second question, CoT decoding can indeed be applied to the RLSF model, followed by a second round of RLSF. This approach yields a significant improvement, with accuracy increasing by at least 10 percentage points in the second round. However, the gains from further iterations diminish progressively, making additional rounds impractical unless supplemented by new training examples to enrich the model's learning.

---

> > ### Comment · Reviewer_Cz7g · 2024-12-02
> >
> > I appreciate the response from the authors. To clarify my point on the correlation between calibration and accuracy, I think we are on the same page that Phi-2 may not be a good model in an absolute manner, as you mentioned. However, Phi-2 seems to have the best accuracy out of the three models in Table 1, which makes me wonder if the base model's reasoning performance (i.e., accuracy in Table 1) could be the source of gain in RLSF. A more clarification on this point would strengthen the paper's claim.

---

### Official Review · Reviewer_YfRu · 2024-11-03

**Soundness:** 2
**Presentation:** 2
**Contribution:** 2
**Rating:** 3
**Confidence:** 4

**Summary:**

The paper proposes Reinforcement Learning from Self-Feedback (RLSF)to improve logical reasoning in LLMs by utilizing self-confidence scores instead of human feedback for model training. The authors suggest that if a language model is well-calibrated, the confidence in its responses correlates with reasoning quality. They use this self-generated confidence as a reward signal to guide reinforcement learning, aiming to improve model performance on reasoning tasks without relying on human annotations. Experimental results on Multi-Arith and GSM8K show that RLSF-enhanced models outperform baseline models in reasoning tasks.

**Strengths:**

-

**Weaknesses:**

- The paper does not sufficiently explore related work in self-feedback or self-improving methods, such as CoT reasoning or majority-voting-based preference learning. It also lacks a comparison with these baseline methods, which could help clarify the novelty and advantage of their approach.
- The methodology section confusing and lacks specific details on the proposed model's implementation. Figure 1 depicts an inconsistency: it shows PPO as the optimization technique, while the experiments utilize DPO, introducing ambiguity regarding the methods used.
- The experimental analysis lacks sufficient depth. The paper does not demonstrate the superiority of the self-feedback-based reward model over simpler baseline methods, such as majority voting. Additionally, the high Expected Calibration Error (ECE) in the reward model suggests potential limitations in the model's capability for RL tasks, which the paper does not adequately address.
- The proposed method is only validated on Phi-2 model. Experiments on stronger models like Phi-3, llama 3 would be helpful to demonstrate the generalization of RLSF.

**Questions:**

-The reward model's ECE appears high, raising concerns about the reliability of self-confidence for RL training. Could the authors address whether such a reward model is effective for reinforcement learning or provide improvements to better calibrate it?
- How to determinate if the models used in the experiments is well-calibrated? If not, can it be used for RL training, as this is the assumption of the RLSF?

---

> ### Author Response · Authors · 2024-11-29
>
> Thank you for your insightful feedback and critique. Below, we address your comments point by point to clarify and enhance the understanding of our work:
>
> Comparison with Related Methods:
>
> We are happy to include CoT reasoning and majority voting-based preference learning as baselines. While we agree these are valuable baselines, they do not fully align with the self-feedback paradigm central to our approach. Specifically, CoT reasoning relies on external prompts or examples.
>
> PPO and DPO in Figure 1
>
> Thank you for pointing out the potential ambiguity in Figure 1. Please note that PPO is our primary optimization method, while DPO is included as a comparative baseline due to its relevance to similar optimization problems.
>
> Expected Calibration Error (ECE)
>
> To clarify, the ECE metrics reported in our paper pertain to the LLMs themselves, not the reward model.
>
> Experiments on Other Models
>
> We share your interest in applying RLSF to stronger models. However, the lack of calibration in models such as Llama-3 currently limits their suitability for our method. We will include results in the revised manuscript but found that when using Gemma, for example, the CoT decoding performance and RLSF performance on the Multi-Arith task is less than 30% accuracy. The model with greedy decoding achieves over 70% accuracy, which illustrates that the poor calibration is harmful to RLSF.

---

### Note · Authors · 2024-12-12

I have read and agree with the venue's withdrawal policy on behalf of myself and my co-authors.